# Lack of the Ig cell adhesion molecule BT-IgSF (IgSF11) induced behavioral changes in the open maze, water maze and resident intruder test

**Dirk Montag**[1]*, **Laura Pelz**[2¤], **Fritz G. Rathjen**[2]

1 Neurogenetics Laboratory, Leibniz Institute for Neurobiology, Magdeburg, Germany, 2 Max-Delbrück-Center for Molecular Medicine, Berlin, Germany

¤ Current address: National Center for Tumor Diseases (NCT) Heidelberg and the German Cancer Research Center (DKFZ), Heidelberg, Germany

* dirk.montag@lin-magdeburg.de

**Data Availability Statement:** All relevant data are within the paper and its Supporting Information files.

## Abstract

The brain- and testis-specific Ig superfamily protein (BT-IgSF, also termed IgSF11) is a homotypic cell adhesion protein. In the nervous system, BT-IgSF regulates the stability of AMPA receptors in the membrane of cultured hippocampal neurons, modulates the connectivity of chandelier cells and controls gap junction-mediated astrocyte-astrocyte communication. Here, we performed behavioral tests in BT-IgSF-deficient mice. BT-IgSF-deficient mice were similar to control littermates with respect to their reflexes, motor coordination and gating, and associative learning. However, BT-IgSF-deficient mice displayed an increased tendency to stay in the central illuminated areas in the open field and O-Maze paradigms suggesting reduced anxiety or increased scotophobia (fear of darkness). Although BT-IgSF-deficient mice initially found the platform in the water maze their behavior was compromised when the platform was moved, indicating reduced behavioral flexibility. This deficit was overcome by longer training to improve their spatial memory. Furthermore, male BT-IgSF-deficient mice displayed increased aggression towards an intruder. Our results show that specific behaviors are modified by the lack of BT-IgSF and demonstrate a contribution of BT-IgSF to network functions.

## Introduction

Initially BT-IgSF (brain- and testis-specific Ig superfamily protein, also designated IgSF11) was described as a novel IgSF member that is preferentially expressed in the brain and testis BT-IgSF [1]. It functions as a homotypic cell adhesion protein [2, 3] and shares an identical overall domain organization with an N-terminal located V- and a C2-type domain and a highly related amino acid sequence with CAR (coxsackievirus and adenovirus receptor), endothelial cell-selective adhesion molecule (ESAM), and CLMP (CAR-like membrane protein) [4]. Independently, BT-IgSF was also found to be up-regulated in intestinal-type gastric

**Funding:** The authors received no specific funding for this work.

**Competing interests:** The authors declare that no competing interests exist.

cancers and designated IgSF11 [5] (moreover it was also termed V-set and Immunoglobulin domain containing 3, abbreviated VSIG-3) [3, 6]. The cytoplasmic segment of BT-IgSF contains a PDZ-recognition motif that binds to scaffolding proteins such as PSD95 [7].

So far, the function of BT-IgSF has been studied in neural cells, Sertoli and germ cells of the testes, during osteoclast differentiation and in the organization of pigment cells in fish [2, 7–14]. In the murine testis BT-IgSF is found in Sertoli cells at the blood-testis barrier, a structure that opens and closes to allow the passage of germ cells. In a global knockout the absence of BT-IgSF causes a malfunction of the blood-testes-barrier due to a mislocalization of connexin43, which was found throughout the seminiferous epithelium instead of being restricted to the blood-testes barrier [9]. In line with this finding connexin43 has been shown to play an essential role in the reassembly of the blood-testes barrier [15]. A critical role of BT-IgSF in regulating the organization of pigment cells into stripes along the dorso-ventral or anterior-posterior body axes was observed in zebrafish and *Neolamprologus meeli* [2, 11, 12]. Similar irregular patterns of chromatophores have been described in zebrafish with mutations in *connexin 41.8* and *connexin 39.4* suggesting a functional link between BT-IgSF and connexins [16–19].

In the brain, BT-IgSF is strongly expressed in GFAP-positive astrocytes, ependymal cells and more weakly in neurons [14]. Knockdown studies using cultured mouse hippocampal neurons indicated that BT-IgSF (IgSF11) is implicated in synaptic transmission and plasticity through a tripartite interaction with PSD95 and AMPA receptors [7]. In this *in vitro* experiment, knockdown of BT-IgSF (IgSF11) led to the increased mobility of AMPA receptors and resulted in enhanced endocytosis of AMPA receptors. Therefore, BT-IgSF might be important for the stabilization of AMPA receptors in neuronal plasma membranes [7]. In another study on neurons, BT-IgSF was found to regulate the innervation of pyramidal neuron axon initial segments by chandelier cell axons [8]. More recently, we demonstrated that the absence of BT-IgSF perturbed gap-junction mediated communication between astrocytes in the hippocampus and cortex as shown by biocytin spread. In addition, we found that BT-IgSF-deficient mice displayed a reduction in their expression level of astrocytic connexin43 protein and an impaired clustering of connexin43 [14].

Overall, BT-IgSF expression in the brain may modify the information flow in neuronal or astrocyte networks and as a consequence might affect the behavior. Therefore, we investigated the behavior of $BT\text{-}IgSF^{-/-}$ mice using different paradigms. Here, we show that lack of BT-IgSF results in distinct changes in open mazes, spatial orientation and increased male aggression.

## Materials and methods

### Ethics statement

Animal experiments were in line with the guidelines for the welfare of experimental animals and approved by the local authorities of Sachsen-Anhalt and Berlin (LaGeSO numbers T0313/97, X9007/16, O 0038/08) and carried out in accordance with the European Communities Council Directive of 24th November 1986 (86/609/EEC) and directive 2010/63/EU of the European Parliament on the protection of animals used for scientific purposes.

### Mice

BT-IgSF-deficient mice (B6-*Igsf11*$^{tm1e(KOMP)Wtsi/FGR}$) and genotyping have been described elsewhere [9]. Heterozygous mice were intercrossed to obtain homozygous mutants and wild type (WT) littermates that served as control. Animals were housed on a 12/12 h light/dark cycle with free access to food and water.

## Behavior

For the behavioral analysis, sex- and age-matched wild type littermate mice were used as control. During the light phase (12h/12h light-dark cycle), mice were subjected to a series of behavioral tests by an experimenter not aware of the genotype. The behavior of three independent cohorts of mice was investigated (cohort #1: 6 KO males and 7 WT males; cohort #2: 5 KO males and 6 WT males; cohort #3 6 KO males 6 WT males 6 KO females 6WT females) for each sex and genotype. First, mice were housed singly for at least two weeks and then a resident intruder test as described by [20] was conducted. Followed by a series of tests as described in [21–23]. Briefly, after assessment of general parameters indicative of the health and neurological state of the mice, the following behavioral paradigms were conducted in sequential order:

For the *resident intruder test*, a foreign C57BL/6NCrl age- and weight-matched male was placed into the home cage of the male test mouse. The latencies to first contact and first attack and the number and duration of attacks were determined from video records. The test was terminated after 240 sec.

*Grip strength* was measured with a high-precision force sensor to evaluate neuromuscular functioning (TSE Systems GmbH, Bad Homburg, Germany).

*Rota-rod performance*. Animals received two training sessions (3 h interval) on a rota-rod apparatus (TSE) with increasing speed from 4 to 40 rpm for 5 min. After 4 days, mice were tested at 16, 24, 32, and 40 rpm constant speed. The latency to fall off the rod was measured.

*Open field*. Exploration was assessed by placing mice in the middle of a 50 x 50 cm arena for 15 min. Using the VideoMot 2 system (TSE), tracks were analyzed for path length, visits, walking speed, and relative time spent in the central area (infield), in the area close to the walls (<10 cm, outfield), and in the corners.

*O-Maze*. Mice were placed in the center of an open area of an O-maze (San Diego Instruments). Their behavior during 5 min was recorded on videotape. Number of entries into the closed or open areas was counted and the time spent in these compartments was determined using the VideoMot 2 system (TSE).

*Light-dark Avoidance*. Anxiety-related behavior was tested by placing mice in a brightly lit compartment (250 lux, 25 x 25 cm) adjacent to a dark compartment (12.5 x 25 cm). The number of transitions between the compartments and the time spent within each were analyzed during 10 min. As a test for long-term memory [24] animals were placed at the last day of testing again in the light-dark avoidance box. The latency to enter the dark compartment was measured and compared to the latency at the first time in the box.

*Acoustic startle response and prepulse inhibition (PPI)*. A startle stimulus (50 ms, 120 dB) was delivered to the mice in a startle-box system (TSE) with or without preceding prepulse stimulus (30 ms, 100 ms before the startle stimulus) at eight different intensities (73–94 dB, 3 dB increments) on a 70 dB white noise background. After habituation to the box (3 min), 2 startle trials were followed in pseudo-random order by 10 startle trials and 5 trials at each of the prepulse intensities with stochastically varied intertrial intervals (5–30 s). The maximal startle amplitude was measured by a sensor platform.

Associative learning was assessed by *two-way active avoidance* in a two-chambered shuttlebox (TSE) with 10 s light as conditioning (CS) and electrical foot-shock as unconditioned stimulus (US, 5 s, 0.5 mA pulsed) delivered after the CS (80 trials/ day, 5–15 s stochastically varied intertrial intervals, 5 consecutive days) using mice of cohorts #1 and #3. Compartment changes during CS were counted as conditioned avoidance reactions.

For *conditioned fear testing*, the experimental protocol used by [25–27] was followed closely. Mice of cohort #2, which were not used in the two-way active avoidance shuttle box, were

trained and tested on 2 consecutive days. Training consisted of placing the subject in an operant chamber (San Diego Instruments) and allowing exploration for 2 min. Afterwards, an auditory cue was presented for 15 sec with a footshock (1.5 mA un-pulsed) delivered during the last 2 sec of the auditory cue. This procedure was repeated, and mice were returned to the home cage 30 sec later. 24 hours after training, mice were returned to the same chamber in which training had occurred (context), and freezing behavior (immobility) was recorded. At the end of the 5 min context test, mice were returned to their home cage. One hour later, mice were placed in a novel environment and freezing behavior (immobility) was recorded for 3 min. The auditory cue (CS) was then presented for 3 min and freezing behavior (immobility) was recorded. Freezing scores are expressed as percentage for each portion of the test.

To address spatial learning, we used 3 different *Water Maze* protocols. The water maze consisted of a dark-grey circular basin (130 cm diameter) filled with water (24–26˚C, 30 cm deep) made opaque by the addition of white paint. A circular platform (10 cm diameter) was placed 1.5 cm below the water surface. Protocol 1 was used for cohorts #1and #2 in which mice were submitted to 6 trials per day for 5 days. They were allowed to swim until they found the platform or 120s had elapsed. In this last case, animals were guided to the platform and allowed to rest for 10s. The hidden platform remained at a fixed position for the first 3 days (18 trials, aquisition phase) and was moved into the opposite quadrant for the 2 last days (12 trials, reversal phase). Trials 19 and 20 were defined as probe trials to analyze the precision of the spatial learning. All trials were videotaped and analyzed using the VideoMot 2 system (TSE) and the Wintrack analysis software (release 2000). Protocol 2 was used for cohort #2 after all other tests were finished. Mice were submitted to 4 trials per day for 2 days. They were allowed to swim until they found the platform or 60s had elapsed. The hidden platform was placed at the same position as during acquisition after protocol 1 for the first 8 trials, followed by a trial without platform (probe trial 1) and was then moved into the opposite quadrant for 2 more days (8 trials, reversal), and finally followed by a trial without platform (probe trial 2). Protocol 3 was used on cohort #3 and lasted 10 days with 4 trials (maximum 60 sec) per day for 2 days, followed by a trial without platform (probe trial 1), again 4 trials per day for 2 days and probe trial 2, and again 4 trials per day for 2 days and probe trial 3. Afterwards the hidden platform was placed at a different position every day (opposite, left, right quadrant, and center) for 4 days with 4 trials followed by a daily probe trial.

## Statistical analysis

Behavioral data were analyzed using analysis of variance (ANOVA with genotype and sex as factors) and *post hoc* analysis using Scheffe's test (STATVIEW Program, SAS Institute Inc., Cary, NC) to determine statistical significance. For the rota-rod, open field, and startle/PPI experiments, statistical analysis was additionally performed using repeated measures ANOVA (with between-subject factor genotype and within-subject factor session). A *P*-value smaller than 0.05 (p<0.05) was considered significant.

## Results and discussion

### Behavioral analysis of BT-IgSF-deficient mice

To investigate potential behavioral deficits in mice lacking BT-IgSF, BT-IgSF-deficient mice (KO) and wild type littermates (WT) were subjected to a series of behavioral tests. During the neurological examination, transgenic mice did not display obvious abnormalities with respect to body posture, reflexes (uprighting, eye-blink), or general sensory perception (vision, hearing, touch, pain). As expected, body weight differed significantly between male and female mice (2-way ANOVA $F_{sex(1,44)} = 21.263$, p<0.0001), but not between genotypes (KO males

(n = 17) 28.0±0.7 g, WT males (n = 19) 29.2±0.7 g; KO females (n = 6) 24.1±0.4 g, WT females 24.8±0.9 g). Likewise, the maximum and average grip strength were significantly higher for males compared to females (2-way ANOVA $F_{sex(1,20)}$ = 12.794, p = 0.0019; $F_{sex(1,20)}$ = 12.228, p = 0.0023, respectively, cohort #3 24 mice n = 6 for each sex and genotype), but did not differ between BT-IgSF-deficient and control mice. Furthermore, motor coordination examined on the Rota-Rod was similar for BT-IgSF-deficient and control mice (Fig 1A), indicating that coordination is not generally impaired in adult BT-IgSF-deficient mice.

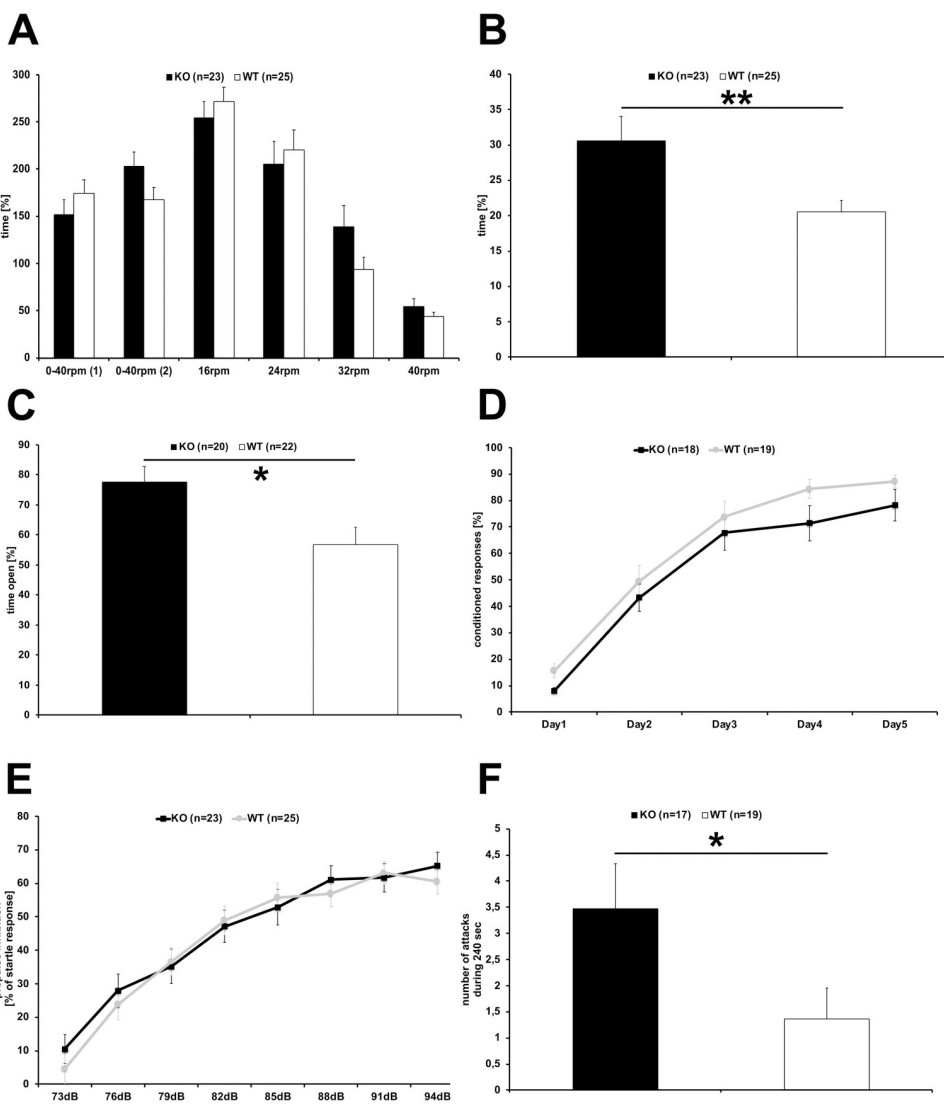

**Fig 1. Behavioral assessment of BT-IgSF-deficient mice.** BT-IgSF-deficient mice and control wildtype littermates were subjected to a series of behavioral paradigms. Motor coordination tested on the rotarod (A) did not differ between BT-IgSF-deficient (black columns) and control mice (white columns). In the open field (B) BT-IgSF-deficient mice (black column) stayed significantly longer in the center compared to their wild-type littermates (white column). In the O-Maze (C) BT-IgSF-deficient mice (black columns) spend more time in the open areas compared to littermate controls (white column). In the two-way active avoidance shuttle box paradigm (D) BT-IgSF-deficient (black line) and control mice (grey line) showed similar acquisition of the task as measured by conditioned responses indicating normal associative learning. Prepulse inhibition of the startle response (E) was not affected in BT-IgSF-deficient mice (black squares) in comparison to wild type littermates (grey bullets). Male BT-IgSF-deficient mice (black column) displayed increased aggression in the intruder test (F) compared to controls (white column) measured by the number of attacks.

In the open field test, BT-IgSF-deficient mice spent significantly more time in the center compared to control littermates (Fig 1B; 1-way ANOVA $F_{(1,46)}$ = 7.374, p = 0.0093; KO (n = 23) 275.0±30.9 sec, WT (n = 25) 185.2±14.1 sec). Furthermore, the distance travelled by BT-IgSF-deficient mice in the wall regions of the maze was significantly shorter (1-way ANOVA $F_{(1,46)}$ = 7.448, p = 0.009, KO (n = 23) 44.6±3.0 m; WT (n = 25) 54.7±1.9 m). These behavioral differences may indicate reduced anxiety, increased scotophobia, or increased explorative activity in BT-IgSF-deficient animals.

In the light-dark avoidance paradigm, BT-IgSF-deficient mice (n = 22) made fewer transitions (14.7±2.1) between compartments, spent more time (338.6±36.5 sec) in the illuminated compartment, and entered the dark compartment after longer latency (207.6±37.3 sec) compared to WT mice (n = 25, 19.9 ±2.6 transitions; illuminated 283.8±30.5 sec; latency 183.6 ±33.9 sec). However, these differences were not statistically significant. When tested for memory, BT-IgSF-deficient mice and control littermates both showed reduced latencies to enter the dark compartment in comparison to the first exposure to the box, which indicates formation of long-term memories independent of the genotype. Still, BT-IgSF-deficient mice took longer to enter the dark compartment when compared to control littermates (BT-IgSF-deficient 167.7±50.4 sec; WT 118.0±39.2 sec, not significant).

In the O-Maze, BT-IgSF-deficient mice spent significantly longer time in the open areas (Fig 1C, 1-way ANOVA $F_{(1,40)}$ = 7.205, p = 0.0105, KO (n = 20) 77.7±5.1% of time, WT (n = 22) 56.7±5.8% of time) and moved similar distances in the open, but less in the closed areas as compared to control littermates (distance closed 1-way ANOVA $F_{(1,40)}$ = 6.565, p = 0.0143, KO (n = 20) 4.44±0.98m, WT (n = 22) 8.18±1.07m). In summary, BT-IgSF knockout mice spend more time in the open illuminated areas in these mazes.

In the two-way active avoidance shuttle box paradigm, the learning curve as measured by the number of conditioned responses was similar for BT-IgSF-deficient and WT control mice (Fig 1D) indicating that associative learning e.g., the association of the light signal and the foot-shock was not compromised in BT-IgSF-deficient mice.

When their startle response and pre-pulse inhibition was analyzed, BT-IgSF-deficient mice displayed a slightly but not significantly reduced startle response at 120 dB (1-way ANOVA $F_{(1,46)}$ = 2.997, p = 0.0901) which was inhibited by pre-pulses of intensities between 73 and 94 dB similar as for WT mice (Fig 1E). These results support that auditory perception, sensorimotor gating and processing are not altered in BT-IgSF-deficient mice.

Fear conditioning was analyzed using mice from cohort #2 mice that were not subjected to the shuttle box. No significant differences between BT-IgSF-deficient (n = 5) and WT (n = 6) male mice with respect to freezing in the context or in the neutral environment with or without tone were detected (). Likewise, context memory (% freezing context—% freezing neutral) and tone memory (% freezing with tone in neutral—% freezing neutral without tone) were not significantly different in BT-IgSF-deficient compared to control mice. These data support the normal formation of associative memories observed in the shuttle box test.

BT-IgSF-deficient mice were further investigated in the Morris Water maze for the acquisition and flexibility of their spatial memory. First, cohort #1 and 2 were subjected to water maze protocol 1 with acquisition for 18 trials over 3 days (D1-3), followed by moving the position of the platform for the next 12 trials over 2 days (D4-5). BT-IgSF-deficient mice displayed similar acquisition (D1-3) as WT mice with respect to escape latency (Fig 2A) and path length (Fig 2B). After platform reversal (D4-5) BT-IgSF-deficient mice differed significantly from WT showing longer escape latencies caused by longer times of immobility (Fig 2C). To further investigate the flexibility of their spatial memory, cohort #2 was subjected to water maze paradigm 2 (Fig 2D–2G). Acquisition during 8 trials (D1-2) was followed by 1 probe trial without platform. For the reversal, the platform was moved to a new position for 8 trials (D3-4)

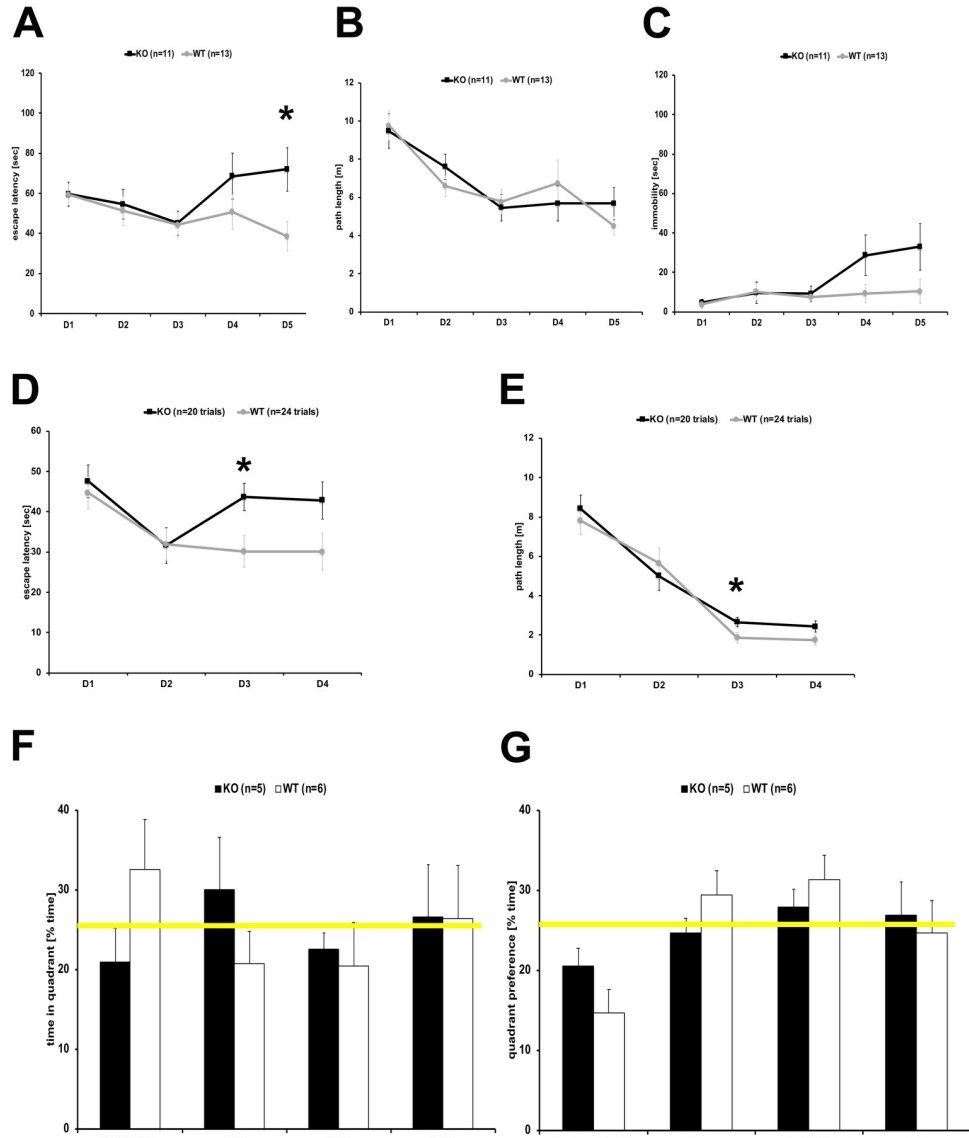

**Fig 2. Spatial learning of BT-IgSF-deficient mice in the water maze.** In the water maze protocols 1 (A-C) and 2 (D-G) with short acquisition phase, BT-IgSF-deficient (black line) showed similar escape latencies (A, D) and pathlengths (B, E) during acquisition but needed much longer time after the platform was moved to a different position compared to littermate controls (grey line). This was mainly caused by longer times of immobility (C). Whereas control mice (white columns) established a goal quadrant preference (F) quickly, BT-IgSF-deficient mice (black columns) did not establish a significant preference for the goal quadrants (F, G). Yellow line indicates chance level 25% in F (probe trial 1) and G (probe trial 2).

followed by a second probe trial without the platform. Again, BT-IgSF-deficient mice showed longer escape latencies but similar path length after replacing the platform (Fig 2D and 2E). Notably, WT but not BT-IgSF-deficient mice established a quadrant preference (>25% of time) for the goal quadrant displayed in probe trial 1 (Fig 2F). In probe trial 2, the preference for the new goal quadrant was similar for BT-IgSF-deficient and wild type mice (Fig 2G). To investigate their spatial learning further, a third water maze protocol was conducted with cohort #3 (Fig 3). Escape latencies (Fig 3A) and path length (Fig 3B) were similar for WT and BT-IgSF-deficient mice. However, in this protocol the establishment of goal quadrant

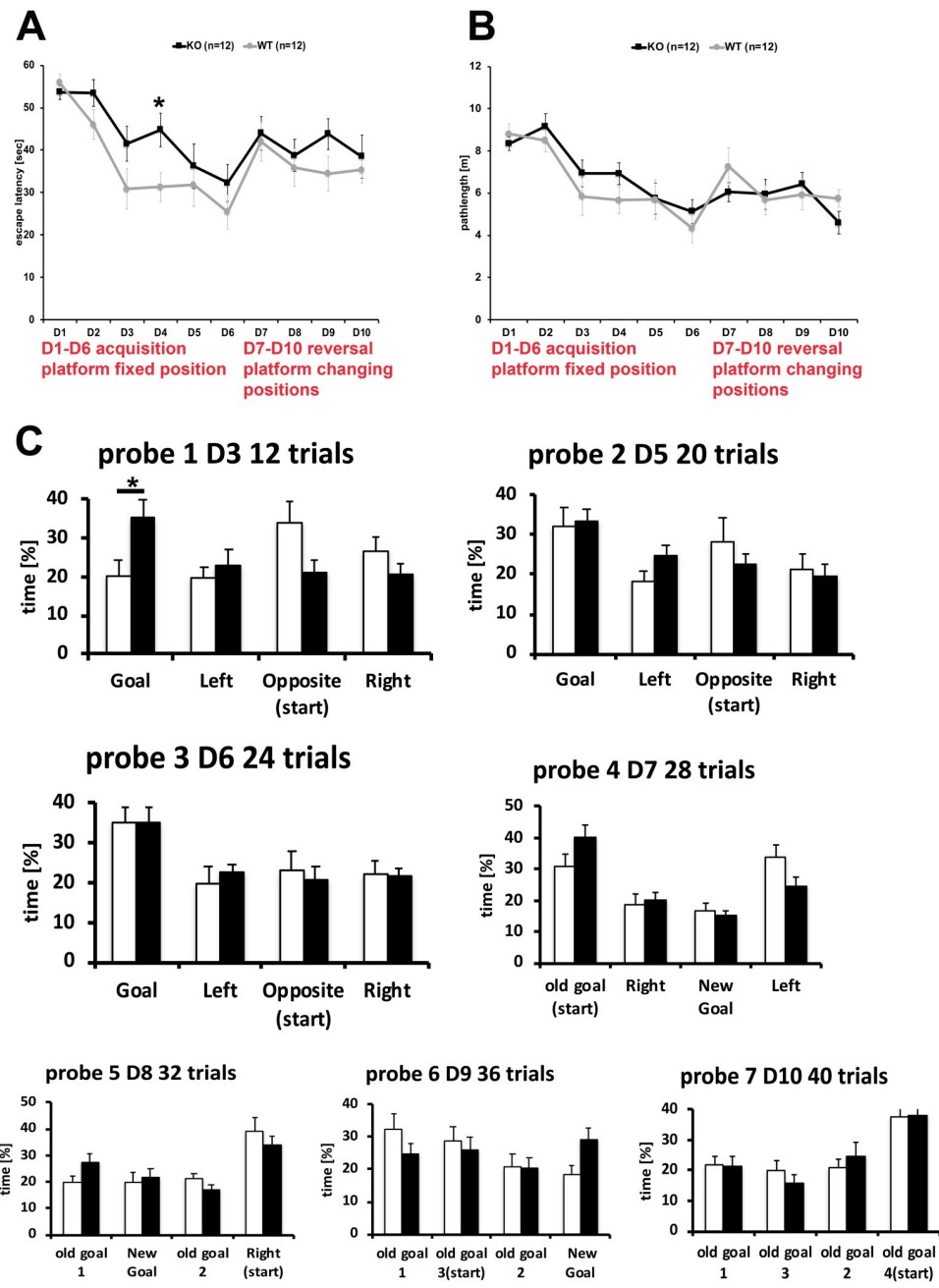

**Fig 3. Flexibility in spatial learning of BT-IgSF-deficient mice in the water maze.** To investigate, whether longer training improved spatial learning in BT-IgSF-deficient mice, the water maze protocol 3 was conducted. During this long acquisition phase and reversals, the escape latencies (A) and pathlengths (B) of BT-IgSF-deficient (black line) and control littermate mice (grey line) did not differ significantly contrasting the compromised flexibility after short acquisition (Fig 2). The evaluation of the goal quadrant preferences during the interspersed probe trials (C) revealed that control littermate mice (white columns) had established a preference already after 12 trials (probe 1) whereas BT-IgSF-deficient (black columns) required 20 trials (probe 3). After changing the platform position daily, quadrant preferences of BT-IgSF-deficient and controls were similar (probe 4–7).

preferences was used an indicator for spatial learning and was assessed by 3 probe trials (probe 1–3) during the acquisition phase and 1 probe trial after each platform position shift (probe 4–7) (Fig 3C). WT mice established a goal quadrant preference after 12 trails (probe 1),

whereas BT-IgSF-deficient mice required 20 trials (probe 2). Only after 24 trials (probe 3), BT-IgSF-deficient mice displayed similar preference and quadrant distinction as WT mice. During the reversal trials, escape latency, path length, and quadrant preferences for WT and BT-IgSF-deficient mice were similar. Therefore, after a longer 6 day training and pronounced acquisition period, BT-IgSF-deficient mice mastered the task similarly well as their control littermates. In conclusion, BT-IgSF-deficient mice required more training to establish a robust spatial memory and displayed subtle deficits in the flexibility of the spatial memory when challenged with changing platform positions.

During handling of male BT-IgSF-deficient mice, a slightly increased aggressive behavior was observed. Therefore, male BT-IgSF-deficient mice were subjected to a resident intruder test which revealed a significantly increased number of attacks by BT-IgSF-deficient mice compared to WT (1-way ANOVA $F_{(1,34)}$ = 4.178, p = 0.0488; KO n = 17, 3.5±0.9 attacks, WT n = 19, 1.4±0.6 attacks, Fig 1F) accompanied by shorter latency to attack (KO 142.7±19.6 sec; WT 185.1±18.3 sec).

## Conclusions

In the present study, we identified significant behavioral differences between BT-IgSF-deficient and control mice. In several paradigms, we observed that BT-IgSF-deficient mice prefer central illuminated areas compared to controls. This may be interpreted as an indication of reduced anxiety, or alternatively as an increased fear of the dark, or altered explorative behavior. Furthermore, we identified a reduced flexibility to adjust for changes in the platform location in water maze paradigms and observed increased aggressive behavior in BT-IgSF-deficient mice.

BT-IgSF is expressed by neurons and astrocytes in the developing, as well as mature nervous system [7, 14, 28]. Functional studies on BT-IgSF (IgSF11) in neurons showed that it regulates the localization of AMPA receptors. Binding of BT-IgSF to the scaffolding protein PSD95 ensures its localization to excitatory neurons that also stabilizes AMPAR receptors at synapses. Knockdown of BT-IgSF mRNA in neuronal cultures increased AMPA receptor mobility in neurons as measured by high-throughput single molecule tracking. Consequently, BT-IgSF deletion in mice resulted in a suppression of AMPA receptor-mediated synaptic transmission in the dentate gyrus and long-term potentiation in the CA1 region of the hippocampus [7]. Furthermore, functional studies on BT-IgSF in astrocytes indicated a severe defect in gap junction mediated astrocyte-astrocyte coupling, while a global BT-IgSF knockout led to an increase in connexin43 clustering and a decrease in dye spread *via* gap junctions in the cortex and hippocampus. These data indicate that BT-IgSF is essential for correct gap junction-mediated astrocyte-astrocyte communication [14]. Overall, we conclude that BT-IgSF expression in neurons and astrocytes may interfere with information flow in neuronal or astrocyte networks and as a consequence might modulate the network properties influencing behavioral performance. However, the contribution of each of these communication defects measured in neurons or in astrocytes lacking BT-IgSF to behavior remains to be determined by future research.

The observed aggressive behavior might be due to change in the hormonal balance in the hypothalamic-pituitary axis in the absence of BT-IgSF. However, differences in blood levels of luteinizing hormone (LH), follicle-stimulating hormone (FSH), or testosterone were not detected between wild type and BT-IgSF-deficient males [9]. Interestingly, AMPA receptors have been implicated in aggressive and social behaviors [29, 30]. Mice deficient for the GluR1 subunit-containing AMPA receptors display reduced intermale aggression [31]. As already pointed out above, the synaptic stabilization of AMPARs is promoted through the dual

interaction of PSD95 with BT-IgSF (IgSF11) and AMPARs. The loss of BT-IgSF in mice results in decreased excitatory synaptic strength in dentate gyrus granule cells and decreased long-term potentiation at SC-CA1 synapses in the hippocampus [7]. The synaptic stabilization of AMPARs is promoted by the interaction of IgSF11 with PSD-95 and AMPARs and loss of IgSF11 in mice results in decreased excitatory synaptic strength in DG granule cells and decreased LTP at SC-CA1 synapses in the hippocampus [7]. Therefore, the increased aggression of BT-IgSF-deficient mice towards an intruder may be related to altered AMPA receptor activity. Indirectly, perturbed gap-junction mediated communication between astrocytes in the hippocampus and cortex of BT-IgSF-deficient mice may interfere with neuronal network properties influencing behavioral performance. Furthermore, traits of aggression and anger are often observed in patients with major depressive disorder, dementia, and other neuropsychiatric disorders [32–34]. However, to our knowledge, mutations in the BT-IgSF gene have not been associated with any of these conditions. Disturbed emotional regulation, such as abnormally low levels of anxiety, indicated by the behavior of BT-IgSF-deficient mice in open spaces may result in or reflect excessive aggression. An overall interpretation of the phenotype of BT-IgSF-deficient mice may indicate the reduced ability to quickly adjust to changes in the environment.

## Supporting information

**S1 Data.**
(XLSX)

**S2 Data.**
(XLSX)

**S3 Data.**
(XLSX)

**S4 Data.**
(XLSX)

## Acknowledgments

The authors gratefully acknowledge expert technical assistance by Karla Sowa and Karola Bach for mouse breeding. The critical reading of Elijah Löwenstein (MDC) is greatly acknowledged. FGR is an emeritus professor at the MDC. FGR thanks Dr Carmen Birchmeier (MDC, Berlin) for generous support and insightful discussions.

## Author Contributions

**Conceptualization:** Dirk Montag, Fritz G. Rathjen.

**Data curation:** Dirk Montag.

**Formal analysis:** Dirk Montag.

**Investigation:** Dirk Montag, Laura Pelz.

**Project administration:** Fritz G. Rathjen.

**Supervision:** Fritz G. Rathjen.

**Visualization:** Dirk Montag.

**Writing – original draft:** Dirk Montag, Fritz G. Rathjen.

**Writing – review & editing:** Dirk Montag, Fritz G. Rathjen.

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
