## [Decision Letter · Decision Letter 0]

6 Oct 2022

PONE-D-22-21240Lack of the Ig cell adhesion molecule BT-IgSF (IgSF11) induced behavioral changes in the open maze, water maze and resident intruder test

PLOS ONE

Dear Dr. Montag,

Thank you for submitting your manuscript to PLOS ONE. After careful consideration, we feel that it has merit but does not fully meet PLOS ONE’s publication criteria as it currently stands. Therefore, we invite you to submit a revised version of the manuscript that addresses the points raised during the review process.

Address any conflicts between the reviews so that it's clear which advice the authors should followProvide specific feedback from your evaluation of the manuscriptPlease ensure that your decision is justified on PLOS ONE’s publication criteria and not, for example, on novelty or perceived impact.

We look forward to receiving your revised manuscript.

Kind regards,

Jamuna Subramaniam, Ph.D

Academic Editor

PLOS ONE

Journal Requirements:

Reviewers' comments:

Reviewer's Responses to Questions

**Comments to the Author**

1. Is the manuscript technically sound, and do the data support the conclusions?

Reviewer #1: Partly

2. Has the statistical analysis been performed appropriately and rigorously? 

Reviewer #1: Yes

3. Have the authors made all data underlying the findings in their manuscript fully available?

Reviewer #1: Yes

4. Is the manuscript presented in an intelligible fashion and written in standard English?

Reviewer #1: Yes

5. Review Comments to the Author

Reviewer #1: The manuscript by Montag et al., have demonstrated the impact of BT-IgSF in various behaviour paradigms. It is a novel finding and has implications in understanding diseases induced by this particular mutation. They authors conclude by stating that specific behaviours are modified by the lack of BT-IgSF and demonstrate a contribution of BT-IgSF to network functions, which they have not shown it experimentally. The manuscript will be strengthened if they could show the below mentioned points.

1.The authors have mentioned about the role of AMPAR in the behavioural abnormalities they have observed in this mouse model. However, they should have showed using electrophysiology or Immunohistochemistry or immunoblot techniques. Furthermore, given the role of AMPARs in memory formation and lack of it, they should have done one or two experiments on the impact of these mutations on basal synaptic transmissions.

2.The authors extrapolate the results to altered function of gap junction mediated communication between astrocytes and the hippocampus. However, the authors should have demonstrated these hypotheses using IHC.

3.Does this mutation impact sociability in these mice?

6. PLOS authors have the option to publish the peer review history of their article (what does this mean?). If published, this will include your full peer review and any attached files.

Reviewer #1: **Yes: **James Chelliah

---

## [Author Response · Author response to Decision Letter 0]

9 Nov 2022

We thank for the overall positive evaluation of our manuscript.

Referee 1 asked for additional experiments/information on 

 1) the AMPA receptor (electrophysiology or immunohistochemistry) and 

 2) on gap junctional communication between astrocytes (immunohistochemistry)

 in the absence of BT-IgSF

 3) whether this mutation impacts sociability

We agree with referee 1 that such data strongly support our interpretations of the behavioral deficits in the absence of BT-IgSF which are drawn in our manuscript. Indeed, such data on the AMPA receptor including electrophysiological measurements and immunohistochemistry in the absence of BT-IgSF (also termed IgSF11) have already been published by Jang et al. 2015 (ref. #7). Furthermore, studies on gap junctional communication between astrocytes including immunohistochemistry, biochemistry and biocytin spreading in the cortex and hippocampus in the absence of BT-IgSF have been recently published by Pelz et al., 2022 (ref. #14). Therefore, now we have cited and discussed these published data in detail in the conclusion paragraph of the revised version of our manuscript. We feel that these additions improved our manuscript in that they illustrate the contribution of BT-IgSF to network functions in the nervous system. The increased aggression of BT-IgSF11-/- males reported in our manuscript dominates sociability with littermates and their infertility affects interaction with females.

All changes that we added are marked in red in our manuscript file "Revised Manuscript with Track Changes PONE-D-22-21240". All original data are now provided in supplementary files.

We hope very much that with these additions our manuscript is now acceptable for publication in PLOS ONE.

---

## [Editor Report · Decision Letter 1]

21 Dec 2022

Lack of the Ig cell adhesion molecule BT-IgSF (IgSF11) induced behavioral changes in the open maze, water maze and resident intruder test

PONE-D-22-21240R1

Dear Dr. Montag,

We’re pleased to inform you that your manuscript has been judged scientifically suitable for publication and will be formally accepted for publication once it meets all outstanding technical requirements.

Kind regards,

Jamuna Subramaniam, Ph.D

Academic Editor

PLOS ONE

Additional Editor Comments (optional):

I am glad to state that your manuscript is accepted based on the correlation between Bt-IGsF1 deficiency altering behaviors in these mice.
---

## [Editor Report · Acceptance letter]

27 Dec 2022

PONE-D-22-21240R1 

Lack of the Ig cell adhesion molecule BT-IgSF (IgSF11) induced behavioral changes in the open maze, water maze and resident intruder test 

Dear Dr. Montag:

I'm pleased to inform you that your manuscript has been deemed suitable for publication in PLOS ONE. Congratulations! Your manuscript is now with our production department. 

Kind regards, 

on behalf of

Dr. Jamuna Subramaniam 

Academic Editor

PLOS ONE